# The SERM/SERD bazedoxifene disrupts ESR1 helix 12 to overcome acquired hormone resistance in breast cancer cells

Sean W Fanning[1], Rinath Jeselsohn[2,3], Venkatasubramanian Dharmarajan[4], Christopher G Mayne[5], Mostafa Karimi[6], Gilles Buchwalter[2], René Houtman[7], Weiyi Toy[8], Colin E Fowler[1], Ross Han[1], Muriel Lainé[1], Kathryn E Carlson[9], Teresa A Martin[9], Jason Nowak[3], Jerome C Nwachukwu[10], David J Hosfield[1], Sarat Chandarlapaty[8], Emad Tajkhorshid[5], Kendall W Nettles[4], Patrick R Griffin[4], Yang Shen[6], John A Katzenellenbogen[9], Myles Brown[2,3], Geoffrey L Greene[1]*

[1]Ben May Department for Cancer Research, University of Chicago, Chicago, United States; [2]Center for Functional Cancer Epigenetics, Dana-Farber Cancer Institute, Boston, United States; [3]Department of Medical Oncology, Dana-Farber Cancer Institute, Boston, United States; [4]Department of Molecular Medicine, The Scripps Research Institute, Jupiter, United States; [5]Department of Biochemistry, College of Medicine, Center for Biophysics and Computational Biology, University of Illinois at Urbana-Champaign, Urbana, United States; [6]Department of Electrical and Computer Engineering, TEES-AgriLife Center for Bioinformatics and Genomic Systems Engineering, Texas A&M University, Texas, United States; [7]PamGene International BV, 's-Hertogenbosch, The Netherlands; [8]Human Oncology and Pathogenesis Program, Memorial Sloan Kettering Cancer Center, New York, United States; [9]Department of Chemistry, University of Illinois at Urbana-Champaign, Urbana, United States; [10]Department of Integrative Structural and Computational Biology, The Scripps Research Institute, Jupiter, United States

*For correspondence: ggreene@uchicago.edu

**Abstract** Acquired resistance to endocrine therapy remains a significant clinical burden for breast cancer patients. Somatic mutations in the *ESR1* (estrogen receptor alpha (ERα)) gene ligand-binding domain (LBD) represent a recognized mechanism of acquired resistance. Antiestrogens with improved efficacy versus tamoxifen might overcome the resistant phenotype in ER +breast cancers. Bazedoxifene (BZA) is a potent antiestrogen that is clinically approved for use in hormone replacement therapies. We found that BZA possesses improved inhibitory potency against the Y537S and D538G ERα mutants compared to tamoxifen and has additional inhibitory activity in combination with the CDK4/6 inhibitor palbociclib. In addition, comprehensive biophysical and structural biology studies show BZA's selective estrogen receptor degrading (SERD) properties that override the stabilizing effects of the Y537S and D538G ERα mutations.
DOI: https://doi.org/10.7554/eLife.37161.001

## Introduction

Estrogen receptor alpha (ERα) plays critical roles in the etiology, treatment and prevention of the majority of breast cancers (*Frasor et al., 2004*). Due to the high degree of efficacy and wide therapeutic indices of endocrine therapies, patients may receive such treatments for progressive disease over the course of several years (*Toy et al., 2013*). Unfortunately, the majority of ER+ metastatic breast cancers that initially respond to endocrine treatment will become refractory despite continued

ERα expression (*Toy et al., 2013*). Selective estrogen receptor modulators (SERMs) like tamoxifen are antagonistic in the breast and agonistic in the bone and endometrium. SERM agonist activity stems from tissue-specific co-regulator binding in the presence of tamoxifen (*Shang and Brown, 2002*). In addition, somatic mutations to *ESR1* (gene for ERα) ligand binding domain (LBD) were identified in 25–30% of patients who previously received endocrine treatment (*Toy et al., 2013*; *Jeselsohn et al., 2015*; *Jeselsohn et al., 2014*; *Niu et al., 2015*). Y537S and D538G are the two most prevalent mutations, and pre-clinical studies show that these mutations confer hormone-free transcriptional activity and relative resistance to tamoxifen and fulvestrant treatment (*Toy et al., 2013*; *Jeselsohn et al., 2015*; *Jeselsohn et al., 2014*; *Niu et al., 2015*). Both mutants enable constitutive ERα activity by favoring the agonist-like conformation of the receptor activating function-2 (AF-2) surface and significantly reduce hormone and 4-hydroxytamoxifen (the active metabolite of tamoxifen) binding affinities (*Nettles et al., 2008*; *Fanning et al., 2016*).

Endocrine treatments with improved efficacy could potentially overcome resistance engendered by the activating somatic mutants and other mechanisms. In pre-clinical studies, fulvestrant (FULV, a selective estrogen receptor degrader (SERD) and complete antiestrogen) at high concentrations was the only molecule that reduced the Y537S and D538G ERα mutant transcriptional activity to basal levels (*Toy et al., 2013*; *Jeselsohn et al., 2015*; *Jeselsohn et al., 2014*; *Niu et al., 2015*). However, its clinical efficacy is limited by poor solubility and oral bioavailability (*Wardell et al., 2013a*; *van Kruchten et al., 2015*). Consequently, new complete antiestrogens are being examined for their activities in breast cancers harboring Y537S and D538G ERα that all demonstrate improved oral bioavailability and pharmacokinetics, including G1T48, AZD9496, GDC-0927, RAD1901, SAR439859, and LSZ102 (*Wardell et al., 2017*; *De Savi et al., 2015*; *Weir et al., 2016*; *Toy et al., 2017*; *Dickler et al., 2018*; *Wardell et al., 2015b*; *Bihani et al., 2017*; *Tria et al., 2018*; *Shomali et al., 2017*). Other non-traditional ERα degraders including H3B 6545, which covalently binds to the ERα LBD, and an ER PROTAC from Arvinas are currently in development. The side effect profiles and suitability of these new drugs as long-term endocrine therapies remains to be determined (*Rioux et al., 2018*; *Flanagan et al., 2018*).

Here, we explore whether bazedoxifene (BZA), a potent antiestrogen that retains some SERM properties, shows activity against breast cancer cells that express ESR1 somatic mutants. We chose BZA because it has been extensively studied in clinical trials and is well tolerated. BZA was approved a number of years ago for the use in combination with conjugated equine estrogens for hormone replacement therapy in postmenopausal women (DUAVEE, Pfizer) in the US and for the prevention of osteoporosis as a single agent in Europe (*Wardell et al., 2013a*; *Tikoo and Gupta, 2015*). Importantly, it displayed strong antagonist and SERD profiles in the breast while retaining beneficial agonist properties in the bone and did not stimulate endometrial tissue in pre-clinical studies (*Wardell et al., 2013a*; *Komm et al., 2005*; *Lewis-Wambi et al., 2011*). In addition, BZA showed potent antitumor activity in AI/SERM-resistant breast tumors in vivo (*Wardell et al., 2013b*; *Wardell et al., 2015a*). Further, BZA showed good oral bioavailability and improved pharmacokinetics compared with fulvestrant (FULV) (*Wardell et al., 2013a*; *Biskobing, 2007*).

In this study, breast cancer reporter gene assays reveal the inhibitory capacity of BZA against the ERα mutants compared to the SERM 4-hydroxytamoxifen (4-OHT) and SERD FULV in several ER + breast cancer cell lines (MCF-7, ZR75, T47D). We further assessed the ability of BZA to induce the degradation of WT, Y537S, and D538G somatic mutant ERα in MCF7 cells. Additionally, because inhibitors of CDK4/6 combined with antiestrogens are approved for first-line therapy and beyond in metastatic ER+ breast cancers (*Wardell et al., 2015a*; *Dean et al., 2010*; *Yang et al., 2017*), we examined whether the CDK4/6 inhibitor, palbociclib, can be used in combination with BZA to enhance the inhibition of breast cancer cell proliferation. Importantly, comprehensive structural and biophysical studies provide additional molecular insights into the chemical differences between BZA, 4-OHT, and raloxifene (RAL, another SERM) that appear to underlie the SERD properties of BZA and its improved inhibitory efficacy against the Y537S and D538G mutants in breast cancer cells. *Table 1* shows the chemical structures of the molecules examined in this study and summarizes their clinical indications.

**Table 1.** Competitive inhibitors of estrogen receptor alpha.

| Antiestrogen | Class | Approved clinical indications |
|---|---|---|
|  4-Hydroxytamoxifen (4-OHT) | SERM | • Adjuvant treatment for ER + breast cancers (**Early Breast Cancer Trialists' Collaborative Group., 1998**).<br>• Metastatic Breast Cancer (**Lipton, 1982**).<br>• Ductal Carcinoma in Situ (**Allred et al., 2012**).<br>• Reduction in Breast Cancer Incidence in High Risk Women (**Visvanathan, 2009**). |
|  Raloxifene (RAL) | SERM | • Osteoporosis in postmenopausal women (**Messalli and Scaffa, 2010**).<br>• Reduction in Breast Cancer Incidence in High Risk Women (**Cauley et al., 2001**). |
|  Fulvestrant (FULV) | SERD | • First-line therapy for metastatic breast cancer (**Howell et al., 2004**).<br>• Postmenopausal women with progressive breast cancer following other antiestrogen therapy (**Osborne et al., 2002**; **Howell et al., 2002**). |
|  Bazedoxifene (BZA) | SERM/ SERD | • In combination with conjugated equine estrogens (DUAVEE) to prevent postmenopausal osteoporosis (**Tikoo and Gupta, 2015**). |

DOI: https://doi.org/10.7554/eLife.37161.002

## Results

### Bazedoxifene displays SERD activity in MCF-7 cells that express WT ERα

To assess the ability of BZA to inhibit WT ERα in breast cancer cells, we examined its impact on ERα transcriptional activity, degradation and cell growth in MCF-7 cells. 4-hydroxytamoxifen (4-OHT) was used as a representative SERM and FULV was used as a representative SERD (**Figure 1**). In MCF7 cells that expressed an ERE-luciferase reporter gene, BZA was a more potent inhibitor of WT ERα transcription than either 4-OHT or FULV (inhibition of luciferase $IC_{50}$ for BZA = 0.12 nM, 4-OHT = 0.39 nM and FULV = 0.76 nM) (**Figure 1A**). To test the effect of BZA on endogenous WT ERα transcriptional activity, qPCR was used to quantify the relative mRNA levels of known ER target

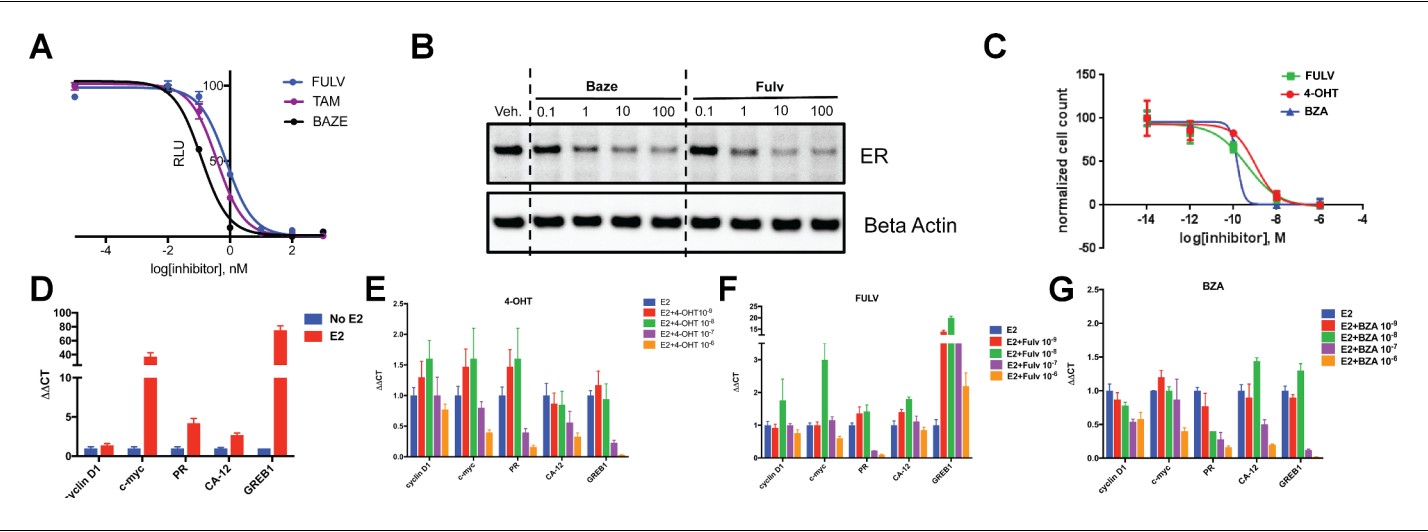

**Figure 1.** The inhibitory potency of BZA in MCF-7 cells. (A) ERα transcriptional reporter gene assay in cells treated with BZA, FULV, and 4-OHT. (B) Relative cyclin D1 mRNA levels of in MCF-7 cells treated with E2 plus 4-OHT, FULV, or BZA vehicle and normalized to E2. (C) Relative c-myc mRNA levelsMCF-7 cells treated with E2 plus 4-OHT, FULV, or BZA vehicle and normalized to E2. (D) Relative GREB1 mRNA levels in MCF-7 cells treated with E2 plus 4-OHT, FULV, or BZA vehicle and normalized to E2. (E) Relative CA12 mRNA levels of MCF-7 cells treated with E2 plus 4-OHT, FULV, or BZA vehicle and normalized to E2. (F) ERα degradation in MCF-7 cells with increasing doses of BZA or FULV normalized to β-actin. (G) Inhibition of cell growth with increasing concentrations of BZA, FULV, or 4-OHT.

DOI: https://doi.org/10.7554/eLife.37161.003

genes, including cyclin D1, c-myc, CA12, and GREB1, in MCF-7 cells treated with estradiol (E2) or with E2 in combination with BZA, 4-OHT or FULV at $10^{-8}$ and $10^{-6}$ M (antagonistic mode). For cyclin D1, 4-OHT increased the mRNA level at $10^{-8}$ M and showed little effect at $10^{-6}$M, while both FULV and BZA decreased mRNA levels at $10^{-6}$ M (*Figure 1B*). The agonist activity of tamoxifen at high concentrations has been described previously (*Horwitz et al., 1978*). BZA increased c-myc mRNA levels at $10^{-8}$ M while it significantly decreased c-myc mRNA at $10^{-6}$ M (*Figure 1C*). Presumably, this effect is similar to the behavior of low-level tamoxifen stimulation and merits further examination. Interestingly, $10^{-6}$ M BZA showed the greatest reduction in mRNA levels of both CA12 and GREB compared to 4-OHT and FULV (*Figure 1D and E*).

As BZA was shown to behave as a SERM/SERD in previous studies (*Wardell et al., 2013a*; *Lewis-Wambi et al., 2011*), we next tested the activity of BZA as an inducer of ERα degradation and observed dose-dependent ER degradation in MCF7 cells. Overall, BZA mediated similar levels of ERα degradation compared to FULV (*Figure 1F*). In terms of cell growth inhibition, BZA showed an improved IC$_{50}$ compared to 4-OHT and in the same range as fulvestrant (BZA IC$_{50}$ = $2.4 \times 10^{-10}$ M, FULV IC$_{50}$ = $3.1 \times 10^{-10}$ M and 4-OHT IC$_{50}$ = $1.19 \times 10^{-9}$ M (*Figure 1G*). Together, these data indicate that BZA degrades WT ERα in breast cancer cells and is more effective at inhibiting ER transcription and cell growth than 4-OHT and FULV.

## BZA is a potent inhibitor of activating somatic mutants of ERα in breast cancer cells

We next tested the activity of BZA in MCF7 cells that ectopically expressed theY537S mutant ERα to determine the inhibition of Y537S mutant cell growth. BZA demonstrated an increased potency compared to FULV and 4-OHT, with an IC$_{50}$ of $1 \times 10^{-10}$ M vs $2 \times 10^{-9}$ M and $7 \times 10^{-9}$ M, respectively (*Figure 2A*). In addition, qPCR data showed that BZA inhibited the transcription of ERα target genes cyclin D1, c-Myc, and PR, in cells expressing the Y537S mutant, in a dose-dependent manner, confirming the on-target effects of BZA in the presence of the mutation (*Figure 2B*).

To evaluate the ability of BZA to induce WT and mutant ERα degradation in breast cancer cells, we treated MCF-7 cells that ectopically expressed HA-tagged WT, Y537S and D538G ERα with BZA and other ligands for comparison. Levels of WT and mutant ERα were quantified using immunoblots with an anti-HA antibody. Cells were treated with 10 nM E2, or 100 nM 4-OHT, 100 nM BZA, 100

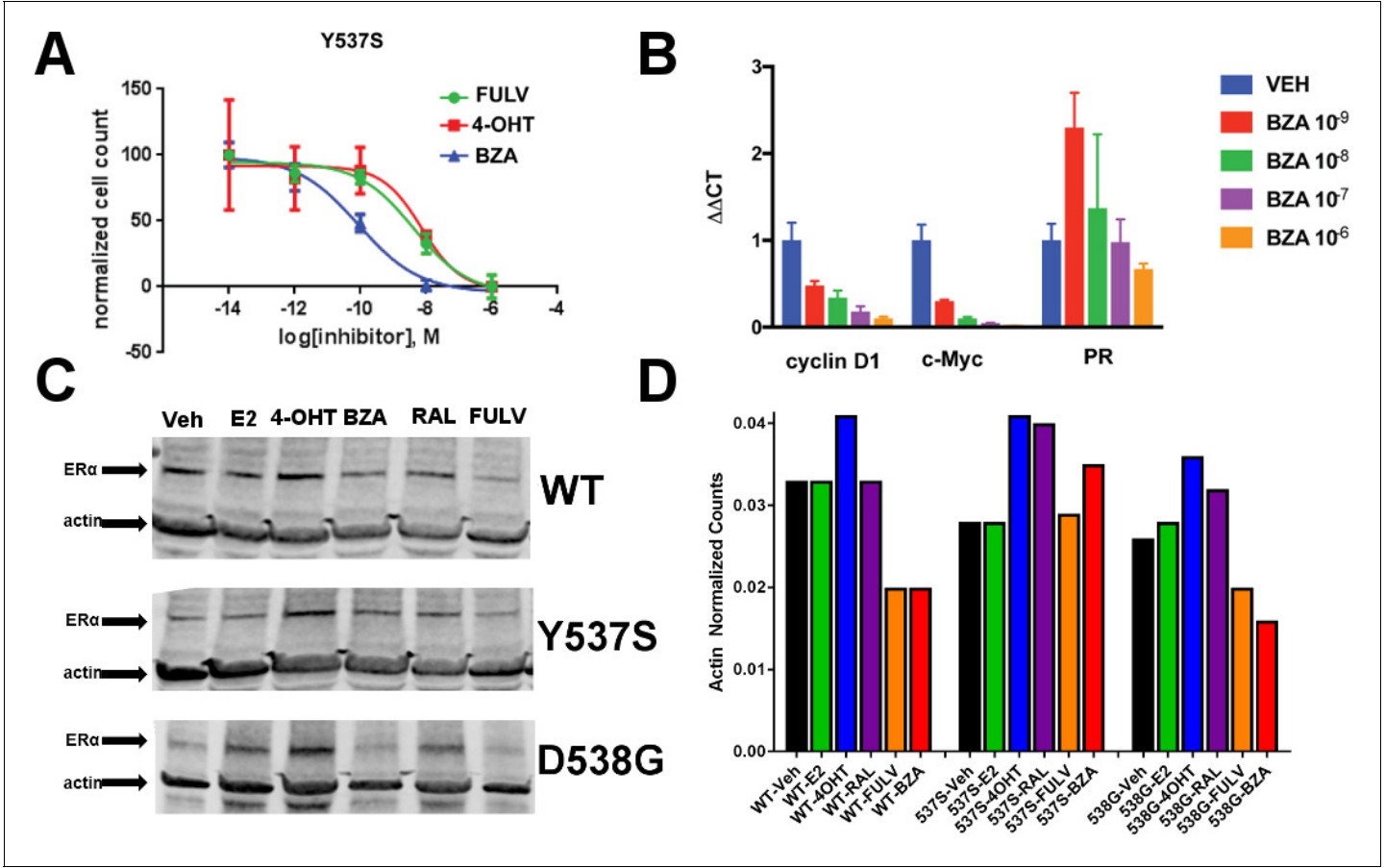

**Figure 2.** The ability of BZA to disrupt Y537S and D538G ERα activity. (A) Cell growth in MCF-7 cells with DOX-induced Y537S ERα expression. (B) Inhibition of ERα target genes in DOX-induced Y537S ERα expressed MCF-7 cells with increasing doses of BZA. (C) Representative immunoblot of HA-ERα WT, 537S, or D538G treated with E2, 4-OHT, BZA, RAL, or FULV for 24 hr. (D) Representative counts of HA-ERα from the immunoblot normalized to actin.

DOI: https://doi.org/10.7554/eLife.37161.006

The following figure supplements are available for figure 2:

**Figure supplement 1.** Replicate experiments of HA-ERα levels in MCF-7 cells upon treatment with E2, 4-OHT, RAL, fulvestrant (FULV), or BZA for 24 hr.
DOI: https://doi.org/10.7554/eLife.37161.004

**Figure supplement 2.** BZA-induced degradation of ERα in CAMA-1, MDA-MB-361, T47D, and ZR75-1 breast cancer cell lines.
DOI: https://doi.org/10.7554/eLife.37161.005

nM RAL, 1 µM FULV or vehicle for 24 hr before immunostaining; 1 µM FULV was chosen because it was the minimal concentration necessary to achieve maximal ERα degradation. All data were normalized to vehicle-treated cells. In cells expressing HA-WT ERα, BZA and FULV induced degradation of the receptor to similar levels while the amount of the receptor increased upon 4-OHT treatment and was slightly reduced with E2 and RAL (*Figure 2C and D*). Interestingly, for the Y537S mutant, ERα expression remained unchanged for E2 and FULV, while it increased for 4-OHT and RAL. Y537S ERα also increased with BZA but less so than RAL or 4-OHT. Surprisingly, BZA degraded the D538G ERα mutant to a greater extent than FULV, while 4-OHT and RAL both increased its expression after 24 hr. It should be noted that BZA and FULV elicited consistent WT and mutant ERα degradation across all replicates *Figure 2—figure supplement 1*. However, 4-OHT and RAL elicited slight variations in the actin-normalized quantity of ERα after 24 hr. Because SERDs can possess differential activities between cell lines (*Guan et al., 2018*), we examined the ability of BZA to induce ERα degradation in T47D, ZR75-1, CAMA-1, and MDA-MB-361 cells between 25 and 100 nM (*Figure 2—figure supplement 2*). In each cell line, near complete degradation was observed at 25 nM BZA. Interestingly, in the T47D cells, an ERα band emerged at 100 nM BZA, suggesting that it is behaving

more SERM-like at that concentration in that cell line. Overall, these data suggest that BZA degrades WT and D538G ERα in MCF7 cells, but that the Y537S mutant is resistant to degradation. However, the levels of Y537S ERα in the BZA-treated cells were still reduced compared to 4-OHT and RAL treatment, consistent with the reduced activities demonstrated by these compounds in MCF-7 reporter gene assays (*Toy et al., 2013*).

## Dual treatment with BZA and palbociclib

CDK4/6 inhibitors have emerged as potent agents in the treatment of metastatic ER+ breast cancer in combination with endocrine treatment. Combined endocrine treatment with a CDK4/6 inhibitor is now the standard of care in either first- or second-line treatment of metastatic ER+ breast cancer (*Wardell et al., 2015a*; *Dean et al., 2010*; *Yang et al., 2017*). Because BZA showed increased activity over FULV and 4-OHT, we explored whether the activity of BZA combined with the CDK4/6 inhibitor, palbociclib (PB), in multiple ER positive cell lines (MCF-7, ZR75, T47D) and long-term, estrogen-deprived (LTED) ER +MCF7 cells that mimic resistance to aromatase inhibitors. For the first three cell lines, the combination of BZA and PB demonstrated the greatest arrest in cellular proliferation, whereas for the LTED cells it was comparable to PB+ FULV (*Figure 3A–D*). Additionally, reduced proliferation of MCF-7 cells expressing the Y537S mutant was observed for the BZA +PB treatment compared to all other treatments (*Figure 3E and F*). Transcriptional reporter gene assays in MCF7 cells showed that: 1. BZA had superior activity in the inhibition of ER transcriptional activity compared to fulvestrant. In addition, palbociclib does not affect ER transcriptional activity either as a single agent or in combination with BZA (*Figure 3G and H*). Similarly, immunoblotting for ER showed that treatment with PB does not affect BZA or FULV-induced degradation of ERα (*Figure 3I*). In sum, these data show that dual inhibition of CDK4/6 with PB and ERα with BZA is an effective combination with significant activity against breast cancer cells expressing WT or constitutively active mutant ERα.

## Coregulator binding specificity and affinities of WT and mutant ERα with BZA, 4-OHT, and FULV

Because hormone regulated coactivator recruitment is crucial for ERα genomic action and inhibition of coactivator recruitment is a key aspect of SERM-mediated ERα antagonism (*Liao et al., 2002*), we tested the effects of 4-OHT, BZA and FULV on co-regulator binding. We applied the Microarray Assay for Real-time Coregulator-Nuclear receptor Interaction (MARCoNI), which allows the quantification of binding affinity of a nuclear receptor with co-regulator peptides. To determine the effect of 4-OHT, BZA and FULV on co-regulator binding to WT, Y537S and D538G ERα, MCF7 cells that ectopically express HA-tagged WT, Y537S, or D538G ERα were used in conjunction with an HA antibody to detect ER binding to the co-regulator array. Experiments were performed under E2 stimulated conditions for WT ER and under apo conditions for mutant ERα. Overall, dose-dependent inhibition of co-regulator binding was observed for the majority of co-regulator peptides with the three drugs (*Figure 4A*). A comparison of $EC_{50}$ levels for inhibition of co-regulator binding of the three ER antagonists showed that $EC_{50}$ levels for FULV in both the WT and mutant cells was higher, as expected given the mechanism of action of FULV, compared to SERMs. The 4-OHT and BZA $EC_{50}$s were higher in the presence of the Y537S and D538G mutations. Collectively, these results show that differences in antagonistic activity manifested by the three endocrine treatments are reflected by changes in co-regulator binding. There are significant differences among these drugs in their antagonistic activity on WT-ER and mutant ER.

## SERMs and SERDs abolish hormone-independent ERα-coactivator binding in vitro and reverse hormone recruitment of coactivators

To further dissect the molecular basis for the reduced BZA, 4-OHT, and RAL potency/efficacy observed with mutant ERs, biochemical coactivator recruitment and competitive ligand-binding experiments were performed. As described previously (*Fanning et al., 2016*), Förster resonance energy transfer (FRET) assays were used to evaluate the interaction of wild-type and Y537S and D538G mutant ERα with steroid receptor coactivator 3 (SRC3), a key coregulator in breast cancer cells. The nuclear receptor recognition domain (NRD) of SRC3 and LBD of the ERs, were used in these experiments.

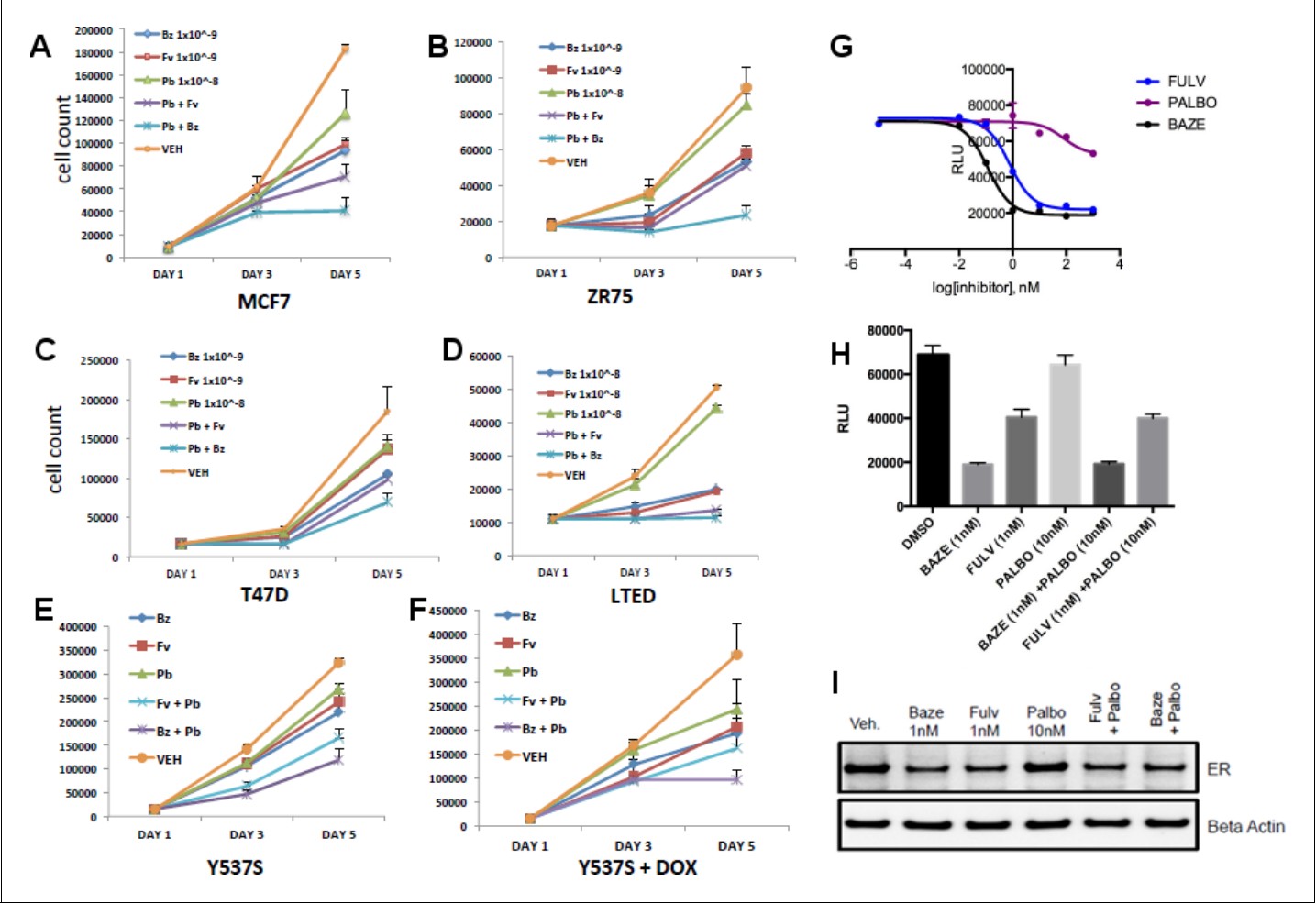

**Figure 3.** Combination treatment with CDK4/6 inhibitor and BZA. (**A**) Cell growth inhibit with MCF7 breast cancer cells. (**B**) ZR75 breast cancer cells. (**C**) T47D breast cancer cells. (**D**) LTED breast cancer cells. (**E**) Non-induced MCF7 breast cancer cells with a dox-inducible ERα Y537S mutant. (**F**) MCF7 cells expressing ERα Y537S. (**G**) Dose-response curves for inhibition of ERα transcriptional activity in the presence of BZA, PALBO, and FULV in MCF7 cells. (**H**) ERα transcriptional reporter gene assays for combination treatments. (**I**) ERα stability resulting from combination treatments.
DOI: https://doi.org/10.7554/eLife.37161.007

Previously, we showed that in the absence of hormone, SRC3 did not bind to WT ERα LBD, whereas Y537S ERα bound SRC3 markedly in the absence of E2 with a 10-fold reduced affinity, and D538G ERα bound SRC3 with a 100-fold reduced affinity compared to hormone-bound WT receptor (*Fanning et al., 2016*). To better ascertain the potency of ligands to inhibit coactivator recruitment, we titrated the ligands into fixed concentrations of LBD and SRC3 and monitored LBD-SRC3 interaction by a FRET assay; the three samples were primed with E2 to get a measurable signal from WT and D538G ER. 4-OHT, BZA and RAL reversed the binding of SRC3 NRD to the two mutant ERs and WT-ERα with similar potencies (*Figure 4B*). Together, these data show that BZA, 4-OHT, and RAL inhibit both the basal and E2-stimulated recruitment of SRC3 coregulator by the WT and mutant ERα in vitro.

## BZA, RAL and FULV elicit similar reduced binding affinities for Y537S and D538G compared to WT ERα LBD

To examine what role alterations in binding affinity may play in this reduced potency, competitive [3H]-E2 ligand binding assays ligand-binding experiments were used to examine the effect of Y537S and D538G mutations on BZA, RAL, and FULV ER binding affinities in vitro. We previously showed that the E2 and 4-OHT binding was significantly reduced for both the Y537S and D538G mutants

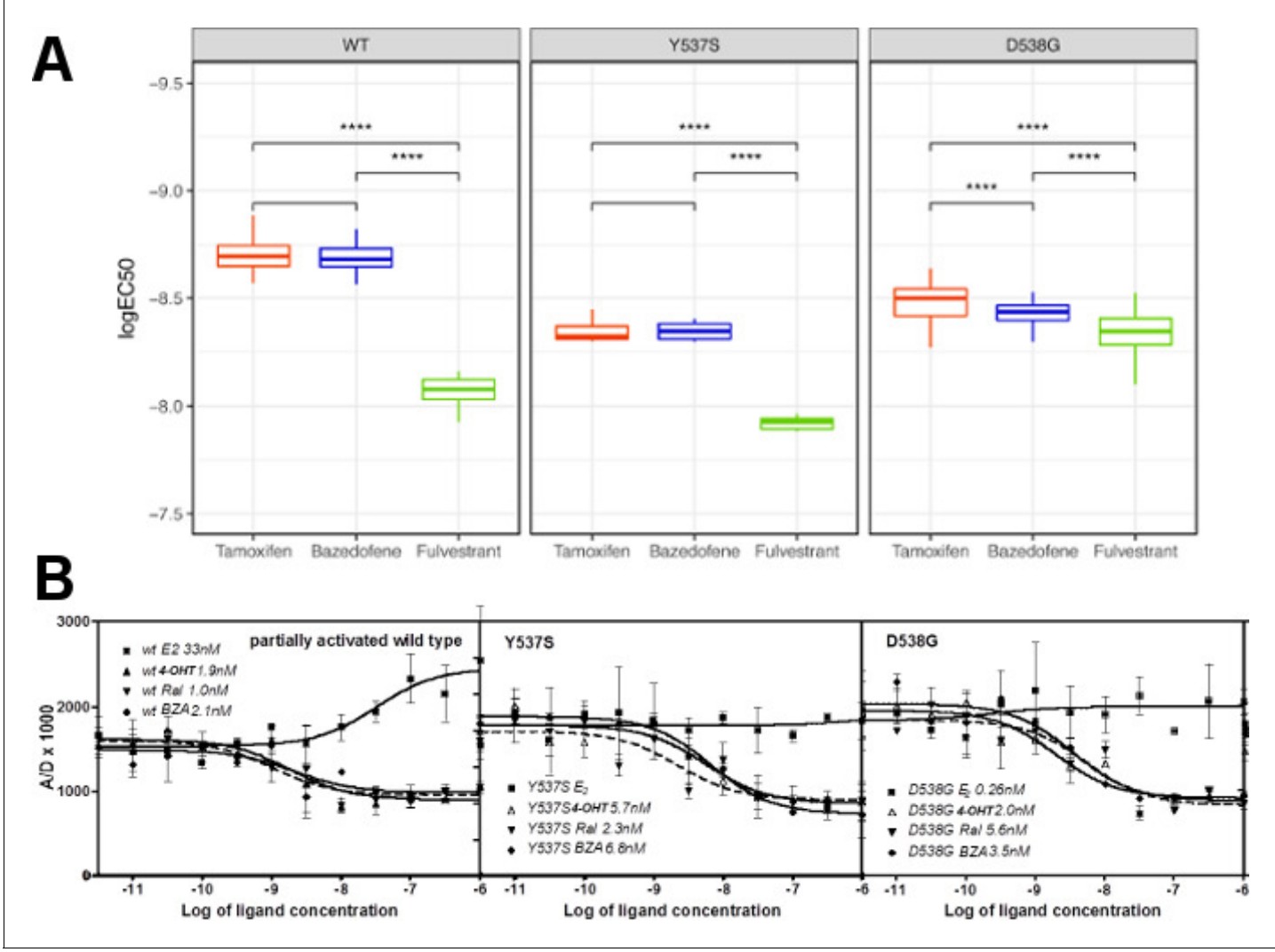

**Figure 4.** Coactivator recruitment and inhibition of WT, Y537S, and D538G ERα LBD. (**A**) EC$_{50}$ quartiles for cells treated with 4-OHT (red), BZA (blue), or FULV (green). (**B**) In vitro quantification of the effect of ligands on promoting (E2) or inhibiting (4-OHT, RAL, BZA) the binding of SRC3-NRD to recombinant expressed WT, Y537S, or D538G ERα LBD. To be able to measure a signal from all three receptors, they were first primed with 20 nME2 before adding ligand. IC$_{50}$ values (nM) are shown next to the legend for each protein.

DOI: https://doi.org/10.7554/eLife.37161.008

(*Fanning et al., 2016*). Affinities were reported using K$_i$ values calculated from IC$_{50}$ values using the Cheng-Prusoff equation (*Table 2*) (*Cheng and Prusoff, 1973*). The affinities of RAL, BZA and FULV for the ERα mutants were reduced 9 to 27-fold relative to WT-ERα. It should be noted that the binding affinity of 4-OHT remained the highest compared to RAL, BZA and FULV against the mutant LBDs. The binding affinities of all tested antiestrogens were somewhat more reduced in D538G than in Y537S. However, FULV and BZA demonstrated the highest potencies in the transcriptional reporter gene assays, even though they exhibited reduced affinities compared to 4-OHT. Therefore, our data indicate that other factors beyond the reduced binding affinity of mutant LBDs for SERM or SERD must play a role in their decreased potency.

## The SERD properties of BZA arise from its disruption of helix 12

X-ray crystallography was used to reveal the structural details of BZA's antiestrogen properties. An X-ray crystal structure for the WT ERα LBD, in complex with BZA was solved to 2.4 Å with two dimers in the asymmetric unit (ASU) (PDB: 4XI3). The BZA ligand and H12 are well resolved in each

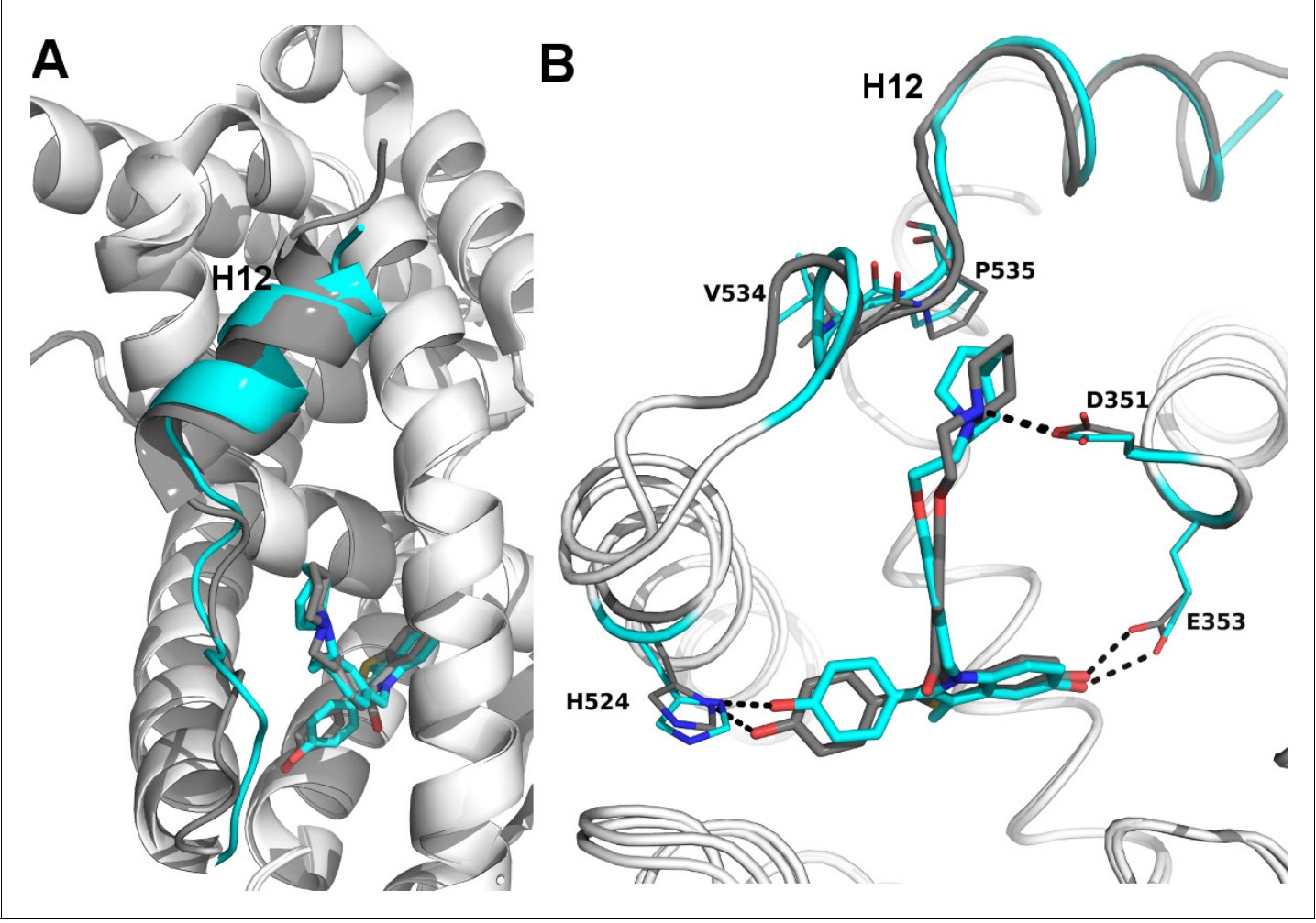

**Figure 5.** Structural basis for the SERD properties exhibited by BZA. (**A**) Overlay of BZA (cyan) with RAL (grey) X-ray crystal structures showing differences in H11-12 loop and H12 orientation. (**B**) Hydrogen bonds (dashed lines) formed by BZA and RAL in the binding pocket and highlighting differences in H11-12 loop and H12 conformation. PDBs: 1ERR and 4XI3.

DOI: https://doi.org/10.7554/eLife.37161.009

The following figure supplement is available for figure 5:

**Figure supplement 1.** 2|Fo-Fc|difference map of BZA in the ERα ligand binding pocket contoured to 1.5σ.

DOI: https://doi.org/10.7554/eLife.37161.010

monomer in the ASU; poorly resolved residues were not included in the model. *Figure 5—figure supplement 1* shows the observed difference map density for the BZA ligand for chain A. We were unable to obtain diffraction quality crystals with BZA or RAL in complex with either the Y537S or D538G mutant LBDs.

Clear structural differences are apparent compared to the previously published molecular modeling and docking simulations of the ERα LBD-BZA complex (*Lewis-Wambi et al., 2011*). Specifically, the C3 methyl on the indole ring of BZA appears to shift the core of the molecule away from M386, L391 and L428 and towards H12. BZA is most structurally similar to RAL; however, BZA displays more SERD-like behavior (*Wardell et al., 2013a*). *Figure 5* shows a superposition of the ERα LBD-BZA or RAL x-ray crystal structures. Interestingly, the distal phenol on BZA appears to form a hydrogen bond with improved binding geometry to H524 compared to the distal phenol of RAL. This suggests that the ketone on the RAL linker constrains the phenyl group, sterically precluding the adoption of an ideal hydrogen bonding geometry with H524. The core differences for BZA broadcast down the linker arm to alter its vector towards H12 where the azepan ring now pushes against

**Figure 6.** HDX MS for WT, Y537S, and D538G ERα LBD in complex with E2. Deuterium uptake for each peptide is calculated as the average of % D for the 6 time points (10 s, 30 s, 60 s, 300 s, 900 s and 3600 s), and the difference in average % D values between the Apo-ERα and ERα-E2 bound samples is shown as a heat map with a color code given at the bottom of the figure (warm colors for deprotection and cool colors for protection). Peptides are colored only if they show a > 5% difference (less or more protection) in average deuterium uptake between the two states, and the software employs a paired two-tailed student's t-test-based coloring scheme (p-value < 0.05 for two consecutive time points or a p-value < 0.01 for a single time point) to distinguish real protection differences from inherent variation in the data. Grey color represents no significant change (0–5%) between the two states.

DOI: https://doi.org/10.7554/eLife.37161.011

The following figure supplements are available for figure 6:

**Figure supplement 1.** HDX MS for WT and D538G ERα LBD in complex with FULV.

DOI: https://doi.org/10.7554/eLife.37161.012

**Figure supplement 2.** HDX MS for WT, Y537S, and D538G ERα LBD in complex with 4-OHT.

DOI: https://doi.org/10.7554/eLife.37161.013

**Figure supplement 3.** HDX MS for WT, Y537S, and D538G ERα LBD in complex with BZA.

DOI: https://doi.org/10.7554/eLife.37161.014

V534 and P535 (*Figure 5B*). The alterations to V534 and P535 propagate to H12 which appears displaced out of the AF-2 cleft into a less stable orientation.

## BZA binding conformation is energetically favored compared to RAL

Quantum mechanical calculations were employed to determine whether inherent differences in the BZA and RAL ligands accounted for differences in antagonist potency. A significant energetic shift was observed between the dihedral scans, revealing that the BZA arm can adopt a set of torsion angles with significantly reduced energetic penalties compared to RAL (*Figure 6A*). Importantly, the energetic minima well is significantly broader for BZA compared to RAL, indicating that the angles adopted by BZA in the ERα ligand-binding site are more favorable than RAL. Furthermore, an energetic penalty of approximately 1 kcal/mol would be incurred by RAL to adopt the same conformation observed for BZA in the X-ray crystal structure. Together these data show that the BZA ligand itself possesses physical properties that are more favorable to impact ERα H12 compared to RAL.

## BZA and FULV reduce the impact of Y537S and D538G mutations on helix 12 dynamics

SERDs competitively bind to the ERα LBD and destabilize helix 12 (H12), leading to proteosomal degradation, while SERMs push the helix into the AF2 cleft to block coregulator binding (*Fanning et al., 2016*). Furthermore, AZD9496, a newer orally available SERD pushes H12 into the AF2 cleft but destabilizes the helix (*De Savi et al., 2015*). Previous work showed that the Y537S and D538G mutants, in complex with 4-OHT, adopts an altered antagonist conformation with respect to the WT-4-OHT complex (*Fanning et al., 2016*). Here, we explored how Y537S and D538G ERα LBD mutations impact H12 mobility in the BZA complex using differential hydrogen-deuterium exchange mass spectrometry (HDX-MS). For comparisons we chose E2 as a representative hormone, FULV as a representative SERD, and 4-OHT as a representative SERM. Average time-dependent amide-deuterium uptake kinetics is indicative of conformational flexibility in proteins with highly dynamic regions being more susceptible to solvent deuterium exchange compared to conformationally rigid regions. As expected, addition of E2 resulted in an increased protection against exchange in H12 (inferred as increased stability or less dynamic), and this protection was enhanced for the Y537S and D538G mutants (*Figure 6*). Also, FULV treatment led to increased $D_2O$ uptake in H12 (interpreted as destabilization) of regions near H12 in both the WT and D538G mutant (*Figure 6—figure supplement 1*), consistent with its SERD-like properties. Unfortunately, we were unable to collect HDX data for the Y537S mutant with FULV because it precipitated out of solution. Similar to previously published data (*Fanning et al., 2016*), binding of 4-OHT resulted in decreased stability of H12 with the Y537S compared to D538G and WT receptor (*Figure 6—figure supplement 2*), suggesting that these mutants resist the ability of the SERM to alter their structure. Interestingly, addition of BZA did not increase the stability (lesser protection) of the region near H12 to as great of an extent as 4-OHT in WT and mutant ERα LBDs, suggesting that BZA-bound ERα adopts a less stable antagonist conformation than 4-OHT-bound ERα (*Figure 6—figure supplement 3*), consistent with the crystal structure. These data suggest that, while their antagonist conformations are altered by Y537S or D538G mutations, the ERα-BZA and FULV complexes maintain potency because they resist the stabilizing impact of the mutations better than 4-OHT.

## How Y537S and D538G erα LBD mutants alter the BZA antagonist structure

To understand the structural basis for the reduced BZA degradation of Y537S compared to WT ERα, atom-level explicit-solvent molecular dynamics (MD) simulations for the LBD of ERα Y537S-BZA and D538G-BZA were performed using the WT-BZA crystal structure as template. When compared to the WT-4-OHT structure (PDB: 3ERT), the D538G-4-OHT structure displayed a significantly altered H11-12 loop, which leads to a perturbed H12 antagonist state (*Figure 6* and *Figure 7B*) (*Fanning et al., 2016*).

MD simulations show that H12 of both BZA-bound mutants remain close to an antagonist conformation (*Figure 7B,D*). Furthermore, H12 conformational fluctuations observed in MD simulations are less pronounced in Y537S-BZA than they are in D538G-BZA (*Figure 7C*), which echoes the aforementioned reduced H12 dynamics observed in HDX-MS data and agrees with our result that Y537S reduces BZA activity more than D538G does. Over the course of the simulations, the Y537S-BZA structure shows a hydrogen bond formation between E380 and S537 (*Figure 7B*), which could contribute to fewer H12 conformational fluctuations in Y537S ERα. It should be noted that this hydrogen

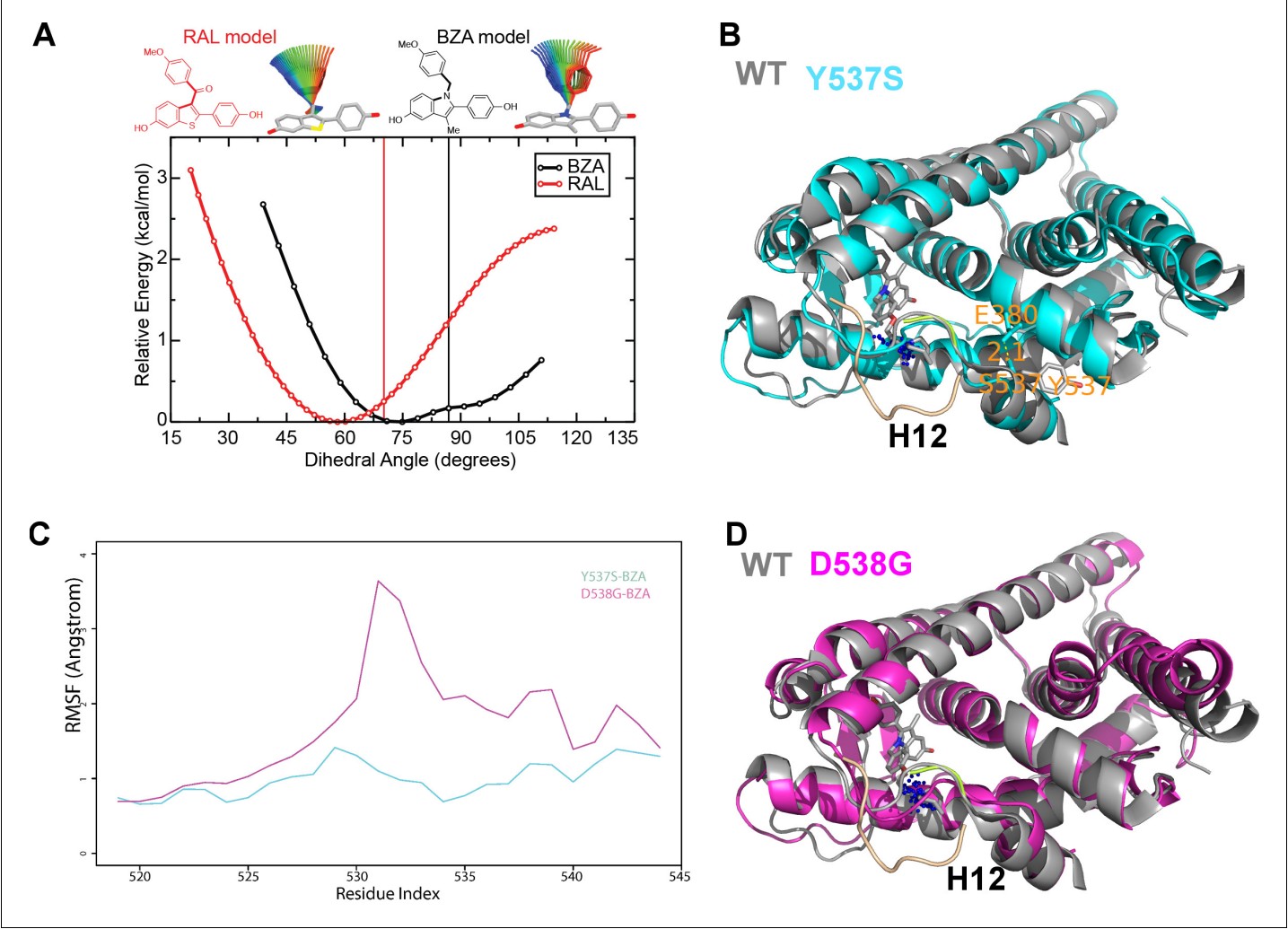

**Figure 7.** (A) QM scan of RAL and BZA arm torsion angle energetics. (B) Overlaid representative calculated structure of Y537S-BZA (cyan) closest to the last 50-ns average from molecular dynamics (MD) simulation and the WT-BZA structure, vertical lines represent the ligand binding conformation in the X-ray crystal structure. Nitrogen atoms of the azepane part of BZA for the last 50-ns MD ensemble are shown in blue spheres near H12. Loop H11-12 is also shown for crystal structures of WT-4-OHT (wheat; PDB: 3ERT) and D538G-4-OHT (yellow; PDB: 4Q50). (C) Root mean squared fluctuations of ERα LBD residues, including the C-end of H11 (resi. 519–527), the loop H11-12 (resi. 528–535), and H12 (resi. 536–544), in the last 50-ns MD simulations for Y537S-BZA (cyan) and D538G-BZA (magenta). (D) Overlay of a representative structure of D538G-BZA (magenta) from MD simulations and the same WT-BZA structure (gray) as in (B).

DOI: https://doi.org/10.7554/eLife.37161.015

bond is not sterically allowed in the WT-BZA crystal structure. This predicted hydrogen bond may stabilize H12 in the AF-2 cleft relative to WT. The simulated H11-12 loop conformations for both Y537S-BZA and D538G-BZA resemble WT-BZA or D538G-4-OHT, being closer to the ligand binding site, than they do to WT-4-OHT. The D538G mutation further increases the conformational variance of the H11-12 loop compared to Y537S (*Figure 7C*), which can be explained by the lack of a hydrogen bond in Y537S and the greater backbone conformational freedom allowed by the mutant glycine residue. Additionally, the H11-12 loop of both Y537S and D538G appears further away from the protein core and BZA compared to that of WT (*Figure 7B,D*). We hypothesize that the varied conformations and increased dynamics of the H11-12 loop in both ERα mutants makes it more difficult for BZA to maintain sufficient interactions with the loop to disrupt the ER antagonist conformation. Together, these data show that both mutations produce a stable antagonist conformation, especially at H12, and reduce the SERD-like properties of BZA by lessening its ability to disrupt the H11-12 loop and stabilize H12 in the AF-2 cleft.

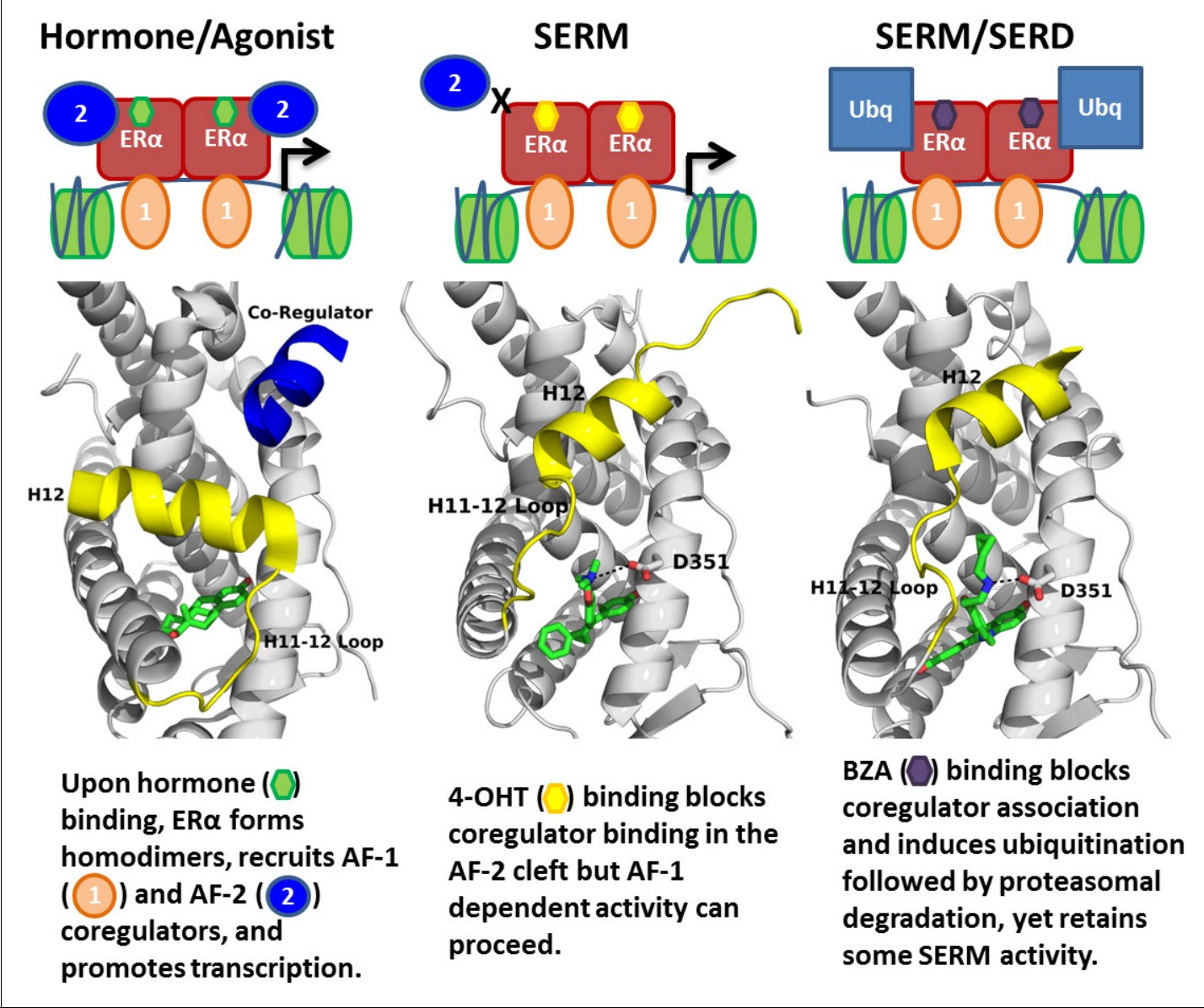

**Figure 8.** Summary of BZA's SERM/SERD activity compared to SERMs and agonist compounds.

DOI: https://doi.org/10.7554/eLife.37161.016

## Discussion

Somatic mutation to *ESR1* following prolonged estrogen-deprivation therapy represents a newly appreciated mechanism of acquired hormone resistance in metastatic breast cancer. The two most prevalent mutations, Y537S and D538G, give rise to a dysfunctional receptor that escapes hormone regulation and has decreased sensitivity to inhibition by 4-OHT and fulvestrant (*Fanning et al., 2016*). Newly characterized orally available pure antiestrogens (e.g. GDC-0927, AZ9496, and RAD1901) are emerging as potentially potent inhibitors of these mutants, but their long-term clinical utility is unknown and their side effect profiles have not been studied in large patient populations (*Toy et al., 2017*; *Lai et al., 2015*). Because BZA is already clinically approved for use in hormone replacement therapy and is a potent ERα antagonist in the breast, an agonist in bone, and neutral in the endometrium with long-term safety data in thousands of patients, we explored its ability to inhibit the Y537S and D538G ERα somatic mutants in breast cancer (*Wardell et al., 2013a*).

We first quantified the ability of BZA to disrupt WT and mutant ERα transcription, induce degradation, and inhibit cell growth in MCF-7 cells. These data show that BZA has increased activity, compared to the SERD FULV and SERM 4-OHT, toward inhibition of cell growth and ERα transcriptional activity. Further, BZA elicited degradation of WT receptor, although not as much as FULV. Importantly, BZA reduced the levels of Y537S ERα relative to RAL and 4-OHT. Together these data suggest that the ability of BZA to degrade ERα confers an inhibitory advantage over 4-OHT in the context of the somatic mutants. Additionally, combined treatment with BZA and the CDK4/6 inhibitor palbociclib resulted in additional inhibitory potency of cellular proliferation in breast cancer cells expressing both WT and Y537S ERα without impacting BZA's action. In line with our studies, previous studies with an Y537S ER mutant PDX model showed an enhanced inhibitory effect on proliferation when bazedoxifene was added to palbociclib (*Wardell et al., 2015a*) As such, combination BZA-CDK4/6 treatment shows significant potential in pre-clinical studies experiments and merits further evaluation. Results from the current study and the aforementioned PDX studies have led to an ongoing Phase Ib/II clinical trial testing the combination bazedoxifene with palbociclib in patients with metastatic ER +breast cancer (NCT02448771). The correlative studies of this trial will include assessment of the *ESR1* mutational status. Although this study does not include an arm with single agent bazedoxifene, the study will provide initial results on the signal of the activity of bazedoxifene in patients that have received prior endocrine treatment.

Comprehensive structural-biochemical investigations were undertaken to understand the basis for the SERD-like properties of BZA and its improved potency against the Y537S and D538G ERα LBD mutants compared to 4-OHT. Biochemical coactivator studies using recombinant ERα LBD demonstrated that BZA could inhibit the both basal and E2-induced recruitment of coregulators in vitro, while ligand-binding assays showed that the Y537S and D538G mutants had significantly reduced affinities for BZA, 4-OHT, RAL, and FULV. Interestingly, an HDX MS comparison of BZA, 4-OHT, and FULV suggested that FULV and BZA can both resist the impact of somatic mutation on their antagonist binding conformations compared to 4-OHT. An X-ray crystal structure of the WT-BZA complex revealed that differences in the core of the molecule translated to an altered vector of the linker arm, resulting in contacts with the H11-12 loop and a perturbed H12 antagonist binding mode. This less stable H12 antagonist conformation likely explains the SERM/SERD properties of BZA, wherein it allows H12 to adopt an antagonist conformation (like a SERM) that is somewhat destabilized, although not as destabilized as when FULV is bound.

Atomistic MD simulations were used to examine the molecular basis for the observed decrease in degradation of ERα within breast cancer cells expressing the Y537S and D538G mutants. Both mutants stabilized the antagonist conformation of H12 in the AF-2 cleft, while the Y537S appeared to be the more stable of the two by forming a hydrogen bond between S537 and E380. Interestingly, E380Q has also been found to be a recurrent ESR1 mutation able to confer endocrine resistance. MD simulations also suggest that these mutants have varied conformations and increased dynamics of loop H11-12, which potentially reduces BZA's ability to disrupt ER antagonist conformations. Together these results suggest that BZA retains its SERM antagonist properties within breast cancer cells expressing Y537S and D538G mutant ERα to a greater extent than 4-OHT but its SERD-like properties are diminished. The reduced potency on the mutants likely derives from the stabilization of the apo receptor in the agonist conformation, which reduces the on rate of ligand binding (*Fanning et al., 2016*), reflecting that the agonist conformer blocks ligand exchange (*Sonoda et al., 2008*).

Overall, our findings show the molecular basis for the SERD-like activity of BZA and its potential advantage with and without CDK4/6 inhibitor, versus 4-OHT, to inhibit the Y537S and D538G ERα mutants. Importantly, interrogating the structural details of BZA-ERα LBD binding suggests that molecules with improved pharmacological profiles that specifically disrupt the H11-12 loop at H12 will have clear advantages against breast cancer cells expressing WT, Y537S, and D538G ERα. *Figure 8* summarizes how BZA achieves SERM/SERD activity compared to SERM or agonist compounds. In fact, the newer SERM/SERDs and SERDs with improved pharmacologic profiles (e.g. AZ9496) appear to do so using a similar mechanism (*De Savi et al., 2015*). Thus, the ability of other new SERDs to withstand the impact of Y537S and D538G mutations on their antagonist-binding modes should be investigated.

# Materials and methods

**Key resources table**

| Reagent type (species) or resource | Designation | Source or reference | Identifier | RRID |
|---|---|---|---|---|
| Recombinant DNA reagent | Estrogen Receptor Alpha Ligand Binding Domain WT or Y537S | Fanning SW, Mayne CG, Dharmarajan V, Carlson KE, Martin TA, Novick SJ, Toy W, Green B, Panchamukhi S, Katzenellenbogen BS, Tajkhorshid E, Griffin PR, Shen Y, Chandarlapaty S, Katzenellenbogen JA, Greene GL. Estrogen receptor alpha somatic mutations Y537S and D538G confer breast cancer endocrine resistance by stabilizing the activating function-2 binding conformation. eLife. 2016;5:e12792. doi: 10.7554/eLife.12792. | | |
| Strain, strain background (Protein Expression) | E.coli BL21(DE3) | https://www.neb.com/products/c2527-bl21de3-competent-e-coli#Product%20Information | | |
| Cell line (Mammary Cells) | MCF7, MCF7 HA-Y537S, MCF7 HA-D538G, T47D, CAMA-1, ZR75-1, MDA-MB-361 | ATCC; this paper; this paper; ATCC; ATCC; ATCC; ATCC | ATCC HTB-22, ATCC HTB-23, ATCC HTB-21, ATCC CRL-1500, ATCC HTB-27 | CVCL_0031, CVCL_0553, CVCL_1115, CVCL_0588, CVCL_0620 |
| Antibody | anti-ERα antibody F10 | Santa Cruz Biotechnology | sc-8002 | AB_627558 |
| Antibody | anti-actin antibody | Santa Cruz Biotechnology | sc-69879 | AB_1119529 |
| Antibody | anti-HA antibody | Cell Signaling | C29F4 | AB_1549585 |
| Software | CCP4i | https://www.ccp4.ac.uk/ccp4i_main.php | | SCR_007255 |
| Software | VMD | https://www.ks.uiuc.edu/Research/vmd/ | | SCR_004905 |
| Software | Gaussion G09 | http://gaussian.com/ | | SCR_014897 |
| Software | GaussView | http://gaussian.com/ | | SCR_014897 |
| Software | HDX Workbench | http://hdxworkbench.com/ | | |
| Software | Bionavigator | PamGene | | |
| Commercial assay or kit | PamStation96 | PamGene | | |
| Commercial assay or kit | MycoAlert Mycoplasma Detection Kit | Lonza | LT07-518 | |

*Continued on next page*

*Continued*

| Reagent type (species) or resource | Designation | Source or reference | Identifier | RRID |
|---|---|---|---|---|
| Chemical compound, drug | protease inhibitor cocktail solution III | CalBiochem | 539134 | |
| Chemical compound, drug | 4-hydroxyt amoxifen | Tocris | 3412 | |
| Chemical compound, drug | ICI 182–780 (fulvestrant) | Tocris | 1047/1 | |
| Chemical compound, drug | Raloxifene | Sigma-Aldrich | R1402 | |
| Chemical compound, drug | Estradiol | Sigma-Aldrich | E8875 | |

## Breast cancer cellular reporter gene assays

Luciferase reporter assay system (Promega) was used to monitor luciferase activity in MCF7 cells with stable expression of ERE luciferase per the manufacturer's recommendations, using a single tube luminometer (BD Monolight 2010). MCF7 cells were plated in six-well plates and treated with increasing doses of BZA, FULV or 4-OHT (0/0.01/0.1/1/10/100/1000 nM) in complete medium for 24 hr. All studies were performed in triplicates, and luciferase results are reported as relative light units (RLU) and normalized with β-Galactosidase activity using Mammalian β-gal assay kit (Thermo Scientific). Mycoplasma was tested every 6 weeks in these cells, and no mycoplasma was detected in the MCF-7 cell lines using the MycoAlert Mycoplasma Detection Kit (Lonza). MCF-7 cells were purchased directly from ATCC and the studies were completed within 1 year of purchase. Cells reached a maximum of 30 passages during these studies.

## Cell proliferation

For proliferation studies the breast cancer cells were plated in 24 well plates ($2.5 \times 10^4$/well). At the indicated time points, the cells were trypsinized. We determined the number of viable cells by Trypan blue exclusion staining and manual counting with a hemacytometer using independent triplicates.

## RT-PCR

Total RNA was isolated with the RNeasy kit (Qiagen). The high-capacity RT kit (Applied Biosystems) was used for preparation of the cDNA, and the PCR reaction was carried out with SYBR Green (Qiagen).

Primers:
c-myc F: 5' –TTCGGGTAGTGGAAAACCAG-3'
c-myc R: 5'-CAGCAGCTCGAATTTCTTCC-3'
PR F: 5'-AGCCAGAGCCCACAATACAG-3'
PR R: 5-'GACCTTACAGCTCCCACAGG-3'
GREB1 F: 5'-CTGCCCCAGAATGGT TTT TA-3'
GREB1 R: 5'-GGACTGCAGAGTCCAGAAGC −3'
CA12 F: 5'-CCACTGTGCTCTGGACAG TTT-3'
CA12 R: 5'- GCCTCTCATGCAACTTCTGG-3'
CCDN1 F: 5'- AACTACCTGGACCGCTTCCT-3'
CCDN1 R: 5'- CCACTTGAGCTTGTTCACCA-3'

## ERα stability immunoblots

Tet-ON MCF7 cells lines (HA-ERα WT, HA-ERα Y537S, HA-ERα D538G) were cultured in DMEM supplemented with 10% fetal bovine serum (FBS), 2 mM L-glutamate, 1% penicillin-streptomycin, 100 μg/mL geniticin, and 100 μg/mL hygromycin. Before western blotting experiments, 300,000 cells were plated in each well of a 6-well culture dish and cultured for 48 hr in DMEM supplemented with 10% charcoal-stripped FBS (SFBS), 2 mM L-glutamate, 1% penicillin-streptomycin, and 0.2 μg/mL

doxycycline. Compounds were diluted in ethanol or DMSO. Cells were treated with either 10 nM estradiol (Sigma), 1 μM FULV (Tocris), 100 nM Ral (Sigma), 100 nM BZA (Pfizer), 100 nM 4-OHT (Tocris), or vehicle (ethanol) for 24 hr. Subsequently, cells were washed twice with 1 mL of ice-cold PBS, harvested via scraping, and pelleted at 4°C. Cells were resuspended in 50 μL lysis buffer [0.1% CHAPS, 50 mM HEPES (pH 8.0), 120 mM NaCl,1mM EDTA, 10 mM Na pyrophosphate, and 10 mM glycerophosphate; supplemented with a protease inhibitor cocktail solution III (CalBiochem)]. Cells were lysed via three freeze-thaw cycles. Lysates were then pelleted at 4°C, and 100 μg of protein was mixed with standard 2x Laemmli buffer. Samples were loaded onto a 10% SDS-polyacrylamide electrophoresis gel, transferred after electrophoresis onto nitrocellulose membrane, and immunoblotted using anti-HA-tag (Cell Signaling C29F4) and anti-actin (Santa Cruz Biotechnology AC-15) antibodies. Membranes were probed using anti-rabbit 800 nm (Rockland) and anti-mouse 680 nm (Rockland) and imaged on the Licor Odyssey. Membrane intensities were normalized to actin control and quantified using ImageStudio. Mycoplasma was tested every 6 weeks in these cells and no mycoplasma was detected in any of the Tet-ON MCF-7 cell lines using the MycoAlert Mycoplasma Detection Kit (Lonza).

For experiments using ZR75-1, MDA-MB-361, CAMA-1, and T47D cells, all cell lines were purchased directly from ATCC. T47D and ZR75-1 cells were grown in RPMI-1640 with 10% FBS. CAMA-1 cells were grown in DMEM with 10% FBS. MDA-MB-361 were grown in L15 media with 20% FBS. Cells were plated at 300 k per well. Once cells reached ~60% confluency, they were placed in charcoal-stripped FBS media for 48 hr. Cells were treated with 0 (vehicle), 25 nM, 50 nM, and 100 nM BZA in DMSO and harvested after 24 hr. Cells were lysed with M-PER lysis reagent. Experiments with CAMA-1 and MDA-MB-361 were performed using traditional Western blot. For those experiments, 100 μg of protein was loaded per lane. Experiments in ZR75-1 and T47D were done using a Wes Automated Western Blot (Protein Simple). For the Wes, 9 μg of protein was loaded per lane. ERα F10 antibody (sc-8002) was used at dilutions of 1:500 for traditional western blotting, and 1:50 for the Wes. Mouse β-actin antibody (60008–1-Ig) was used at dilutions of 1:5000 on the Wes, and rabbit β-actin antibody (20536–1-AP) was used at dilutions of 1:10000 for traditional western blotting.

## Co-regulator interaction profiling

This method has been described previously (Koppen et al., 2009). Cell lysates of MCF7 cells expressing HA-tagged WT-ER, Y537S mutant ER and D538G mutant ER were quantified by ELISA (Active Motif, USA) to enable equimolar input. An array with a set of immobilized peptides representing coregulator-derived NR-binding motifs was incubated with a reaction mixture of crude lysate, vehicle (2% DMSO) with or without 1 μM 17β-estradiol (E2), increasing concentrations of BZA, 4-OHT or FULV (0.1, 1, 10, 100, 1000 nM) and anti-HIS-Alexa488 (Qiagen, USA). Incubation was performed for 40 min at 20°C, followed by removal of unbound receptor by washing and generation of a TIFF image of each array using a PamStation96 (PamGene International). Image processing and quantification of ERα binding to each peptide on the array was performed by Bionavigator software (PamGene International).

## Coactivator binding assays

*Protein preparation for TR-FRET.* Expression, purification, and site-specific labeling of the ERα-LBD was performed as described previously (Tamrazi et al., 2002). Generation of the nuclear receptor interaction domain (NRD) of human SRC3 coactivator has also been described previously (Kim et al., 2005). ERα LBD and the SRC3 fragment were labeled with Mal-dPEG4-biotin (Quanta BioDesign, Powell, OH), and 5-iodoacetamido fluorescein (Molecular Probes, Invitrogen, Eugene, OR), respectively.

## Coactivator recruitment with ligand titration

To raise the background level of SRC3 NRD binding, the ERα LBD preparation (1 nM ER-LBD, 0.25 nM SaTb, 100 nM SRC3-fluorescein) was primed with 20 nM E2, and then increasing ligand concentrations (from $3 \times 10^{-12}$ to $3 \times 10^{-7}$ M) were added. Diffusion-enhanced FRET was determined by a parallel incubation without biotinylated ER-LBD and subtracted as a background signal. The time-resolved Förster resonance energy transfer measurements were performed with a Victor X5 plate

**Table 3.** Crystallographic data collection and refinement statistics.

| | ERα LBD-BZA |
|---|---|
| Data Collection | |
| Space Group | P1 |
| a, b, c (Å) | 53.57, 59.17, 94.14 |
| α, β, γ (°) | 86.76, 75.36, 63.03 |
| Resolution Range | 50–2.49 |
| Number of Reflections | |
| (all/unique) | 63,978/29,080 |
| I/σ (highest resolution) | 1.35 |
| $R_{merge}$ | 8.0 |
| Completeness (%) | 97.6 |
| Redundancy | 2.2 |
| Refinement | |
| Rwork/Rfree | 21.1/29.3 |
| No. Residues/Chain | |
| ERα LBD D538G | 241 |
| GRIP Peptide | 0 |
| Water | 5 |
| Ligand | 1 |
| RMSD | |
| Bond lengths (Å) | 0.010 |
| Bond angles (°) | 1.575 |
| Chiral volume | 0.1016 |
| Ramachandran plot statistics | |
| Preferred number (%) | 837 (97.44%) |
| Additional allowed (%) | 18 (2.10%) |
| Outliers (%) | 4 (0.47%) |

DOI: https://doi.org/10.7554/eLife.37161.017

reader (Perkin Elmer, Shelton, CT) with an excitation filter at 340/10 nm and emission filters for terbium and fluorescein at 495/20 and 520/25 nm, respectively, with a 100-μs delay (*Tamrazi et al., 2005*; *Moore et al., 2015*; *Jeyakumar et al., 2011*). The data, representing 2–3 replicate experiments, each with duplicate points, was analyzed using GraphPad Prism 4, and are expressed as the $IC_{50}$ in nM.

## Ligand binding assays

The dissociation constant, ($K_d$) of estradiol binding to each ER was measured by saturation binding with [$^3$H]17β-$E_2$ and Scatchard plot analysis (*Scatchard, 1949*), as described previously (*Fanning et al., 2016*; *Hurth et al., 2004*). Relative binding affinities (RBA) were determined by a competitive radiometric ligand binding assay with 2 nM [$^3$H]E2 as tracer (*Fanning et al., 2016*). Incubations were at 0°C for 18–24 hr. Hydroxyapatite was used to adsorb the receptor-ligand complex, and unbound ligand was washed away. The determination of the RBA values is reproducible in independent experiments with a CV of 0.3, and the values shown represent the average ±range or SD of two or more separate determinations. $K_i$ was determined by the Cheng-Prusoff equation (*Cheng and Prusoff, 1973*) $K_i = IC_{50}/(1 + [\text{tracer total}/K_d \text{ of tracer}])$.

**Table 2.** Ligand-binding affinities for WT, Y537S, and D538G mutant ERα LBD.

| Ligand/Mutant | $K_d$ (nM)[*] | | |
| --- | --- | --- | --- |
| | WT | Y537S | D538G |
| $E_2$[‡] | 0.22 ± 0.11 | 1.40 ± 0.54 | 1.77 ± 0.66 |
| | | $K_i$ (nM)[†] | |
| 4-OHT[‡] | 0.12 ± 0.003 | 2.64 ± 0.4 | 3.28 ± 0.7 |
| RAL | 0.30 ± 0.05 | 3.59 ± 1.0 | 3.77 ± 1.0 |
| BZA | 0.37 ± 0.01 | 3.50 ± 0.6 | 5.53 ± 0.7 |
| Fulvestrant[‡] | 0.13 ± 0.03 | 3.68 ± 0.8 | 5.06 ± 1.2 |

[*] Measured directly by Scatchard Analysis using [$^3$H]E2.

[†] Calculated using the Cheng-Prusoff equation from the $IC_{50}$ values determined in a competitive radiometric binding analysis using [$^3$H]E2 as a tracer.

[‡] Indicates previously published data (*Fanning et al., 2016*; *Zhao et al., 2017*).

DOI: https://doi.org/10.7554/eLife.37161.018

## Hydrogen/deuterium exchange-mass spectrometry (HDX-MS)

Solution-phase amide HDX experiments were carried out with a fully automated system (CTC HTS PAL, LEAP Technologies, Carrboro, NC; housed inside a 4°C cabinet) as described previously (*Fanning et al., 2016*) with slight modifications.

### Peptide identification

Peptides were identified using tandem MS (MS$^2$ or MS/MS) experiments performed with a LTQ linear ion trap mass spectrometer (LTQ Orbitrap XL with ETD, ThermoFisher, San Jose, CA) over a 70 min gradient. Product ion spectra were acquired in a data-dependent mode and the five most abundant ions were selected for the product ion analysis per scan event. The MS/MS *.raw data files were converted to *.mgf files and then submitted to MASCOT ver2.3 (Matrix Science, London, UK) for peptide identification. The maximum number of missed cleavage was set at four with the mass tolerance for precursor ions ± 0.6 Da and for fragment ions ± 8 ppm. Oxidation to Methionine was selected for variable modification. Pepsin was used for digestion and no specific enzyme was selected in the MASCOT during the search. Peptides included in the peptide set used for HDX detection had a MASCOT score of 20 or greater. The MS/MS MASCOT search was also performed against a decoy (reverse) sequence and false positives were ruled out. The MS/MS spectra of all the peptide ions from the MASCOT search were further manually inspected and only the unique charged ions with the highest MASCOT score were used in estimating the sequence coverage and included in HDX peptide set.

### HDX-MS analysis

For differential HDX experiments, 5 μL of a 10 μM ERα LBD (Apo or in complex with 10-fold excess compound) was diluted to 25 μL with D$_2$O-containing HDX buffer (20 mM Tris-HCl pH 8.0, 150 mM NaCl, 5% glycerol, 1 mM DTT) and incubated at 4°C for 10 s, 30 s, 60 s, 900 s, and 3600 s. Following on-exchange, unwanted forward or backward exchange is minimized, and the protein is denatured by dilution to 50 μL with 0.1% TFA in 5 M urea with 50 mM TCEP (held at 4°C, pH 2.5). Samples are then passed across an immobilized pepsin column (prepared in house) at 50 μL min−1 (0.1% TFA,15°C), and the resulting peptides are trapped onto a C$_8$ trap cartridge (Thermo Fisher, Hypersil Gold). Peptides were then gradient eluted (5% CH$_3$CN to 50% CH$_3$CN, 0.3% formic acid over 6 min, 4°C) across a 1 mm ×50 mm C$_{18}$ analytical column (Hypersil Gold, Thermo Fisher) and electrosprayed directly into a high resolution orbitrap mass spectrometer (LTQ Orbitrap XL with ETD, Thermo Fisher). Percent deuterium exchange values for peptide isotopic envelopes at each time point were calculated and processed using HDX Workbench (*Pascal et al., 2012*). Each HDX experiment was carried out in triplicate with a single preparation of each protein-ligand complex. The intensity weighted mean m/z centroid value of each peptide envelope was calculated and subsequently converted into a percentage of deuterium incorporation. This is accomplished by

determining the observed averages of the undeuterated and using the conventional formula described elsewhere (*Zhang and Smith, 1993*). Statistical significance for the differential HDX data is determined by an unpaired t-test for each time point, a procedure that is integrated into the HDX Workbench software (*Pascal et al., 2012*). Corrections for back-exchange were made on the basis of an estimated 70% deuterium recovery and accounting for 80% final deuterium concentration in the sample (1:5 dilution in $D_2O$ HDX buffer).

## X-ray crystallographic analysis of the WT erα LBD-BZA complex

The 6 × His TEV-tagged ERα-L372S, L536S double mutant LBD was expressed in *E.coli* BL21(DE3) and purified as described (*Sharma et al., 2017*). LBD (10 mg/mL) and incubated with 1 mM BZA overnight at 4°C. LBD-BZA was crystallized using vapor diffusion by hanging drop in 33% PEG 3,350, 100 mM Tris pH 6.6, and 250 mM $MgCl_2$. Diffraction data were collected at the Canadian Light Source at beamline 08ID-1 at a wavelength of 0.97 Å. Indexing, scaling, and structure refinement were performed as described (*Fanning et al., 2016*). *Table 3* shows data collection and refinement statistics. Final coordinates were deposited in the Protein Databank with the accession code 4XI3.

## Quantum mechanical calculations

Torsion scans were performed on the bond connecting the internal substituents to the central core for each ligand. The ligand coordinates were extracted from x-ray crystal structures of BZA (PDB code 4XI3) and RAL (PDB code 2QXS) and all hydrogens were added. Relaxed potential energy surface scans in which the remainder of the structure is geometry optimized at each torsion step were prepared and analyzed using the torsion scan module of the Force Field Toolkit (*Mayne et al., 2013*) (ffTK) plugin of VMD (*Humphrey et al., 1996*). Quantum mechanical calculations were performed using Gaussion G09 (*Frisch et al., 2016*) at the MP2 level of theory with a 6–31G* basis set. Both ligands were scanned using a bidirection technique originating from the crystal structure conformation and scanning outward in the (+) and (-) directions independently. The BZA ligand was scanned in four-degree increments while the RAL ligand required a smaller two-degree step size to avoid discontinuities due to broader conformational changes when taking larger steps.

## Molecular dynamics simulations

*Ligand parameterization.* A 3D structure of BZA (without hydrogen atoms) was first built using the computer program GaussView (version 4.1.2; part of the computer program Gaussian 03 (*Frisch, 2004*). The remaining ligand parameterization was carried out as described (*Fanning et al., 2016*).

*Structure preparation, molecular dynamics, data visualization and analysis.* WT-BZA (PDB: 4XI3) was used as a template to construct starting structures of Y537S-BZA and D538G-BZA. Specifically, chains A and C were chosen among the three dimers in 4XI3 for having the least missing residues in the loop H11-12 region, with ions removed and water molecules retained. Side chain atoms of mutation sites (residue 537 and 538, respectively) were also replaced with the mutant residues. Otherwise, structures were prepared, molecular dynamics were calculated, and data were analyzed/visualized as described (*Fanning et al., 2016*).

## Acknowledgements

Special thanks to Dr. Richard Walter and James Gorin for assistance with structure data collection. The Canadian Light Source is supported by the Canada Foundation for Innovation, Natural Sciences and Engineering Research Council of Canada, the University of Saskatchewan, the Government of Saskatchewan, Western Economic Diversification Canada, the National Research Council Canada, and the Canadian Institutes of Health Research. Research reported in this publication was supported by the Susan G Komen Foundation (SWF), Department of Defense Breakthrough Award (GLG and SC), The Virginia and DK Ludwig Fund for Cancer Research (GLG), National Institute of General Medical Sciences of the National Institutes of Health under Award Number R35GM124952 (YS) and by the National Science Foundation under Award Number CCF-1546278 (YS). The content is solely the responsibility of the authors and does not necessarily represent the official views of the National Institutes of Health or the National Science Foundation. Portions of MD simulations were conducted

with high performance research computing resources provided by Texas A and M University ([http://hprc.tamu.edu/](http://hprc.tamu.edu/)).

# Additional information

## Competing interests

Christopher G Mayne, Gilles Buchwalter: Employee and shareholder of Celgene. René Houtman: Employee of PamGene International. Sarat Chandarlapaty: Dr. Chandaralapaty does not receive financial benefit from any data derived from this publication. However, he has received research funds from Daiichi Sankyo and ad hoc consulting honoraria from Novartis, Sermonix, Context Therapeutics, and Lilly. Geoffrey L Greene: GLG does not have direct financial competing interests in regards to bazedoxifene. However, has received research funds as well as consulting fees from Pfizer and Sermonix Pharmaceuticals and is a member of the scientific advisory board of Olema Pharmaceuticals. The other authors declare that no competing interests exist.

## Funding

| Funder | Grant reference number | Author |
|---|---|---|
| Susan G. Komen | PDF14301382 | Sean W Fanning<br>Geoffrey L Greene |
| U.S. Department of Defense | Breakthrough Award W81XWH-14-1-0360 | Sean W Fanning<br>Weiyi Toy<br>Colin E Fowler<br>Sarat Chandarlapaty<br>Geoffrey L Greene |
| National Cancer Institute | CCSG P30 CA08748 | Sarat Chandarlapaty |
| National Institutes of Health | R01CA204999 | Sarat Chandarlapaty |
| National Institutes of Health | R35GM124952 | Yang Shen |
| National Science Foundation | CCF-1546278 | Yang Shen |
| Breast Cancer Research Foundation | BCRF-17-083 | John A Katzenellenbogen |
| National Institutes of Health | R01CA220284 | John A Katzenellenbogen |
| Virginia and D.K. Ludwig Fund for Cancer Research | | Geoffrey L Greene |

The funders had no role in study design, data collection and interpretation, or the decision to submit the work for publication.

## Author contributions

Sean W Fanning, Conceptualization, Resources, Data curation, Formal analysis, Supervision, Funding acquisition, Validation, Investigation, Visualization, Methodology, Writing—original draft, Project administration, Writing—review and editing; Rinath Jeselsohn, Conceptualization, Data curation, Formal analysis, Funding acquisition, Investigation, Methodology, Writing—original draft, Project administration, Writing—review and editing; Venkatasubramanian Dharmarajan, Christopher G Mayne, Muriel Lainé, Data curation, Methodology, Writing—review and editing; Mostafa Karimi, Data curation, Visualization; Gilles Buchwalter, Weiyi Toy, Data curation, Formal analysis; René Houtman, Data curation, Visualization, Methodology; Colin E Fowler, Ross Han, Data curation, Formal analysis, Methodology; Kathryn E Carlson, Data curation, Formal analysis, Visualization; Teresa A Martin, Data curation; Jason Nowak, Formal analysis, Methodology; Jerome C Nwachukwu, Data curation, Validation; David J Hosfield, Methodology, Writing—review and editing; Sarat Chandarlapaty, Emad Tajkhorshid, Supervision, Writing—review and editing; Kendall W Nettles, Patrick R Griffin, Supervision, Methodology, Writing—review and editing; Yang Shen, Data curation, Validation, Visualization, Methodology; John A Katzenellenbogen, Conceptualization, Visualization, Methodology, Writing—review and editing; Myles Brown, Supervision, Funding acquisition, Writing—review

and editing; Geoffrey L Greene, Conceptualization, Supervision, Funding acquisition, Writing—review and editing

## Author ORCIDs

Sean W Fanning (ID) http://orcid.org/0000-0002-9428-0060
Christopher G Mayne (ID) https://orcid.org/0000-0001-8905-6569
Colin E Fowler (ID) http://orcid.org/0000-0002-6281-2851
Jerome C Nwachukwu (ID) http://orcid.org/0000-0003-4313-9187
Sarat Chandarlapaty (ID) https://orcid.org/0000-0003-4532-8053
Emad Tajkhorshid (ID) http://orcid.org/0000-0001-8434-1010
Patrick R Griffin (ID) http://orcid.org/0000-0002-3404-690X
Yang Shen (ID) http://orcid.org/0000-0002-1703-7796
John A Katzenellenbogen (ID) http://orcid.org/0000-0003-0914-0010
Myles Brown (ID) http://orcid.org/0000-0002-8213-1658
Geoffrey L Greene (ID) http://orcid.org/0000-0001-6894-8728

## Decision letter and Author response

Decision letter https://doi.org/10.7554/eLife.37161.023
Author response https://doi.org/10.7554/eLife.37161.024

## Additional files

### Supplementary files

• Transparent reporting form
DOI: https://doi.org/10.7554/eLife.37161.019

### Data availability

X-ray crystallographic data were deposited in the PDB under the accession code 4XI3.

The following dataset was generated:

| Author(s) | Year | Dataset title | Dataset URL | Database and Identifier |
|-----------|------|---------------|-------------|-------------------------|
| Fanning SW, Mayne CG, Toy W, Carlson K, Green B, Nowak J, Walte R, Panchamukhi S, Tajkhorshid E, Nettles KW, Chandarlapaty S, Katznellenbogen JA, Greene GL | 2015 | Estrogen Receptor Alpha Ligand Binding Domain in Complex with Bazedoxifene | https://www.rcsb.org/structure/4xi3 | Protein Data Bank, 4XI3 |

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
