## [Decision Letter]

Thank you for submitting your article "The SERM/SERD Bazedoxifene Disrupts ESR1 Helix 12 to Overcome Acquired Hormone Resistance in Breast Cancer Cells" for consideration by *eLife*. Your article has been reviewed by two peer reviewers, and the evaluation has been overseen by Charles Sawyers as the Senior and Reviewing Editor. The following individuals involved in review of your submission have agreed to reveal their identity: Jeffrey H Hager (Reviewer #1); Donald McDonnell (Reviewer #2).

After discussing the reviews with one another and the Reviewing Editor, we have drafted this decision to help you prepare a revised submission. All the reviewers are positive about the near-term clinical impact of the work but there is some concern about novelty because some of the findings are predictable/incremental based on prior work with BZE. However, we are willing to consider publication in *eLife* if you can address the main points listed below.

Summary:

The authors report a solid functional and structural characterization of the estrogen receptor modulator Bazedoxifene (BZA) that highlights the potential of BZA as a therapeutic to treat ESR1 mutant bearing hormone-resistant breast cancer. Specifically, the work addresses two main questions: (a) what is the molecular basis for the SERD activity of bazedoxifene? and (b) how do somatic mutations in ER, apparent in advanced disease, impact the pharmacology of BZA? This paper answers both of these questions in a comprehensive manner and highlights the potential clinical utility of bazedoxifene (with a cdk4/6 inhibitor) as a treatment for patients with metastatic ER-mutant positive breast cancer. While the data in the paper is executed and presented well, the manuscript in its current form is a relatively minor advance over what these authors and others have previously published (Lewis-Wambi 2011; Waddell 2013) unless additional data are provided.

Essential revisions:

1) Experiments to benchmark BZA against newer next generation SERDs

As noted in the manuscript, there are multiple, novel, chemically diverse SERDs in clinical development. The paper would be more compelling if there were comparative work describing any potential differences between BZA and the 1 or more next-gen SERDs. Are these agents differentiated from BZA? These data would be of interest to the broader community including the academic and industry scientists who are actively developing novel ER-targeting agents and would lend further support to the ongoing clinical evaluation (NCT02448771) of BZA ± palbo in the setting of acquired hormone resistant breast cancer. In addition, you need to update the readers on the ER-modulators that are in development at the current time for ER-positive breast cancer and mention what is known about their activity on ESR1 mutations (H3B 6545, SAR439859, ARV-ER, RAD1901 G1T48 and LSZ102).

2) Determine BZA degrader activity across a cell line panel

Emerging data indicating that SERDs can exhibit differential "degrader" activity in different ER+ cell lines (Metcalfe AACR 2018). MCF-7 is perhaps the most responsive cell-line to SERD (degrader) activity. To better understand the potential of BZA across the heterogeneity of human tumors, it is important to determine if BZA has broad "degrader" activity by testing in a panel of cell lines.

3) In vivo xenograft data

All of the conclusions are based on in vitro studies. Because of the potential clinical impact, BZA should be evaluated in a xenograft model and benchmarked against other ER degraders such as fulvestrant and RAD-1901 (using dose-regimens that approximate clinical exposures) in ESR1 overexpressing Y537S or D538G (or ideally both) models along with parental MCF-7 as control.

4) Clarification of the current BZA approval and different compositions

Most readers will not understand why the authors mention that BZA is approved as a component of a HT preparation (containing estrogens) and may be confused as to what the authors are proposing as the intervention. Clearly BZA alone can be used in the countries where it is approved as a single agent (most of Europe and Japan). From a regulatory perspective, BZA is not an approved drug in the US. Are the authors suggesting that the antagonist activity of BZA in the Duavee is sufficient to inhibit the activity of the mutants? Given that it is likely that the doses of BZA that will be used for breast cancer treatment are likely to be higher than those available in Duavee this issue needs to be discussed.

---

## [Author Response]

Essential revisions:1) Experiments to benchmark BZA against newer next generation SERDsAs noted in the manuscript, there are multiple, novel, chemically diverse SERDs in clinical development. The paper would be more compelling if there were comparative work describing any potential differences between BZA and the 1 or more next-gen SERDs. Are these agents differentiated from BZA? These data would be of interest to the broader community including the academic and industry scientists who are actively developing novel ER-targeting agents and would lend further support to the ongoing clinical evaluation (NCT02448771) of BZA ± palbo in the setting of acquired hormone resistant breast cancer. In addition, you need to update the readers on the ER-modulators that are in development at the current time for ER-positive breast cancer and mention what is known about their activity on ESR1 mutations (H3B 6545, SAR439859, ARV-ER, RAD1901 G1T48 and LSZ102).

New molecules with citations were added to the Introduction at the end of the second paragraph.

2) Determine BZA degrader activity across a cell line panelEmerging data indicating that SERDs can exhibit differential "degrader" activity in different ER+ cell lines (Metcalfe AACR 2018). MCF-7 is perhaps the most responsive cell-line to SERD (degrader) activity. To better understand the potential of BZA across the heterogeneity of human tumors, it is important to determine if BZA has broad "degrader" activity by testing in a panel of cell lines.

We agree. As such, we have worked over the last few months to determine whether there were any differences in BZA’s ability to induce ER degradation in different cell lines besides MCF7. This took time because each cell line was purchased from ATCC and we needed to determine the best growth and detection conditions. We were able to successfully probe for ERα in CAMA-1, ZR75-1, T47D, and MDA-MB-361. We also tried BT474 but were unable to observe ERα. Cells were treated with BZA between 0 and 100 nM and we observed degradation in each cell line.

Interestingly, the only cell line that showed any differences was T47D, which appeared to stabilize ERα at 100 nM, perhaps suggesting that it is more SERM like for the cell line at the highest concentration. A result that we look forward to investigating further. We have updated the Results and Materials and methods accordingly. We also included a new supplemental figure (Figure 1—figure supplement 2) that shows representative blots for each treated cell line.

*3)* In vivo xenograft dataAll of the conclusions are based on in vitro studies. Because of the potential clinical impact, BZA should be evaluated in a xenograft model and benchmarked against other ER degraders such as fulvestrant and RAD-1901 (using dose-regimens that approximate clinical exposures) in ESR1 overexpressing Y537S or D538G (or ideally both) models along with parental MCF-7 as control.

We did not perform these experiments. The main goal of this work was to determine the molecular-basis for BZA’s potency towards Y537S and D538G ERα. Not only would we learn important insights into its mechanism-of-action but it would help to justify eventual examination in PDX models with the mutants, which come at a substantial cost of time and resources. In addition, Dr. McDonnell’s group already showed that BZA possesses activity in ESR1 mutated PDX models (Wardell 2015). To clarify this, we added sentences citing his paper in the Introduction and Discussion.

4) Clarification of the current BZA approval and different compositionsMost readers will not understand why the authors mention that BZA is approved as a component of a HT preparation (containing estrogens) and may be confused as to what the authors are proposing as the intervention. Clearly BZA alone can be used in the countries where it is approved as a single agent (most of Europe and Japan). From a regulatory perspective, BZA is not an approved drug in the US. Are the authors suggesting that the antagonist activity of BZA in the Duavee is sufficient to inhibit the activity of the mutants? Given that it is likely that the doses of BZA that will be used for breast cancer treatment are likely to be higher than those available in Duavee this issue needs to be discussed.

To clarify this for the reader, we included a few sentences at the end of the Discussion:

“Presumably, a single agent early-phase clinical study of BZA would use increased concentrations compared to its approved formulation in Europe and Japan. As such, future studies using in vivopreclinical models of luminal breast cancers that possess Y537S or D538G ESR1 will need to be completed before the full preclinical potential of BZA/CDK4/6 inhibitor combination can be determined.”